# Graph convolutional networks for learning with few clean and many noisy labels

## Abstract

In this work we consider the problem of learning a classifier from noisy labels when a few clean labeled examples are given. The structure of clean and noisy data is modeled by a graph per class and Graph Convolutional Networks (GCN) are used to predict class relevance of noisy examples. For each class, the GCN is treated as a binary classifier learning to discriminate clean from noisy examples using a weighted binary cross-entropy loss function, and then the GCN-inferred "clean" probability is exploited as a relevance measure. Each noisy example is weighted by its relevance when learning a classifier for the end task. We evaluate our method on an extended version of a few-shot learning problem, where the few clean examples of novel classes are supplemented with additional noisy data. Experimental results show that our GCN-based cleaning process significantly improves the classification accuracy over not cleaning the noisy data and standard few-shot classification where only few clean examples are used. The proposed GCN-based method outperforms the transductive approach (Douze et al., 2018) that is using the same additional data without labels.

## 1 Introduction

State-of-the-art deep learning methods require a large amount of manually labeled data. The need for supervision may be reduced by decoupling representation learning from the end task and/or using additional training data that are unlabeled, weakly labeled (with noisy labels), or belong to different domains or classes. Example approaches are *transfer learning* (Wang & Gupta, 2015), *unsupervised representation learning* (Wang & Gupta, 2015), *semi-supervised learning* (Weston et al., 2008), *learning from noisy labels* (Joulin et al., 2016) and *few-shot learning* (Snell et al., 2017).

*Learning from noisy labels* allows using large-scale data and labels from the web without human annotation effort. Most work focuses on learning the representation jointly with the end task, assuming there is still a considerable amount of clean labeled data (Patrini et al., 2017; Lee et al., 2018; Li et al., 2017). However, for a number of classes only very few or even no clean labeled examples might be available at the representation learning stage. *Few-shot learning* limits the labeled data to very few on the end task, while the representation is learned on a large training set of different classes (Hariharan & Girshick, 2017; Snell et al., 2017; Vinyals et al., 2016). Nevertheless, in many situations, more data with noisy labels are available or can be acquired for the end task.

One interesting mix of few-shot learning with additional large-scale data is the work of Douze et al. (2018), where labels are propagated from few clean labeled examples to a large-scale collection. This collection is unlabeled and actually contains data of many more classes than the end task. Their method overall improves the classification accuracy, but at an additional computational cost; it is a *transductive* method, *i.e.*, instead of learning a parametric classifier, the large-scale collection is still necessary at inference.

In this work, we learn a classifier from few clean labeled examples and additional weakly labeled data, while the representation is learned on different classes, as in few-shot learning. We assume the class names are known, and we use them to search an existing large collection of images with textual description. The result is a set of images with potentially relevant, but noisy labels. As shown in Figure 1, we clean this data using a *graph convolutional network* (GCN) (Kipf & Welling, 2017), which learns to predict a class relevance score per image based on the source (clean *vs.* noisy) of its connections in the graph. Both the clean and the noisy images are then used to learn a classifier,

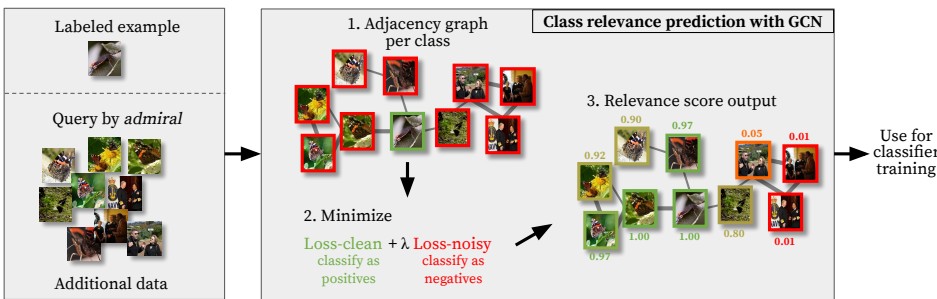

Figure 1: Overview of our cleaning approach for 1-shot learning with noisy examples. We use the class name *admiral* to crawl noisy images from web and create an adjacency graph based on visual similarity. We then assign a relevance score to each noisy example with a graph convolutional network (GCN). Relevance scores are displayed next to the images.

where the noisy examples are weighted by relevance. Unlike most existing work, our method operates independently per class and applies when clean labeled examples are few or even one per class.

We make the following contributions:

- We learn a classifier on a *large-scale weakly-labeled* collection jointly with only *few clean labeled* examples.
- To our knowledge, we are the first to use a GCN to clean noisy data: we cast a GCN as a *binary classifier* learning to discriminate clean from noisy data, and we use its inferred probabilities for the "clean" class as a relevance score per example.
- We apply our method to a few-shot learning benchmark and show significant improvement in accuracy, while outperforming the method by Douze et al. (2018) using the same large-scale collection of data.

## 2   RELATED WORK

**Learning with noisy labels** is often concerned with estimating or learning a *transition matrix* (Natarajan et al., 2013; Patrini et al., 2017; Sukhbaatar et al., 2014) or *knowledge graph* (Li et al., 2017) between labels and correcting the loss function, which does not apply in our case since the classes in the noisy data are unknown. Most recent work on learning from large-scale weakly-labeled data focuses on learning the *representation e.g.* by *metric learning* (Lee et al., 2018; Wang et al., 2018a), *bootstrapping* (Reed et al., 2015), or *distillation* (Li et al., 2017). In our case however, since the clean labeled examples are few, we need to keep the representation mostly fixed.

*Dealing with the noise*, *e.g.* by thresholding (Lee et al., 2018), outlier detection (Wang et al., 2018a) or *reweighting* (Liu & Tao, 2015), is applicable while the representation is learned, based *e.g.* on the gradient of the loss (Ren et al., 2018b). In contrast, the relatively-shallow GCN that we propose effectively decouples reweighting from both representation learning and classifier learning. *Learning to clean* the noisy labels (Veit et al., 2017) typically assumes adequate human verified labels for training, which again is not the case in this work.

**Few-shot learning.** *Meta-learning* (Vilalta & Drissi, 2002) refers to learning at two levels, where generic knowledge is acquired before adapting to more specific tasks. In few-shot learning, this translates to learning on a set of *base classes* how to learn from few examples on a distinct set of *novel classes* without overfitting. For instance, *optimization meta-learning* (Finn et al., 2017; 2018; Ravi & Larochelle, 2017) amounts to learning a model that is easy to fine-tune in few steps. In our work, we study an extension of few-shot learning where more data are available on novel classes, reducing the risk of overfitting when fine-tuning the model. *Metric learning* approaches learn how to compare queries for instance to few examples (Vinyals et al., 2016) or to the corresponding *class prototypes* (Snell et al., 2017). Hariharan & Girshick (2017) and Wang et al. (2018b) learn how to *generate* novel-class examples, which is not needed when more data are actually available.

Gidaris & Komodakis (2018) learn on base classes a simpler cosine similarity-based parametric classifier, or simply *cosine classifier*, without meta-learning. The same classifier has been introduced independently by Qi et al. (2018), who further fine-tune the network, assuming access to the base

class training set. A recent survey (Chen et al., 2019) confirms the superiority of the cosine classifier to previous work including meta-learning (Finn et al., 2017). We use the cosine classifier in this work, both for base and novel classes. All of the above use only the few labeled examples of the novel classes.

Making use of *unlabeled data* has been little explored in few-shot learning until recently. Ren et al. (2018a) introduce a semi-supervised few-shot classification task, where some labels are unknown. Liu et al. (2019) follow the same semi-supervised setup, but use graph-based *label propagation* (LP) (Zhou et al., 2003a) for classification and consider jointly all test images. These methods assume a meta-learning scenario, where only small-scale data is available at each training episode; arguably, such a small amount of data limits the representation adaptation and generalization to unseen data. Similarly, Rohrbach et al. (2013) use label propagation in a *transductive* setting, but at a larger scale assuming that all examples come from a set of known classes. Douze et al. (2018) extend to even larger scale, leveraging 100M unlabeled images in a graph without using additional text information. We focus on the latter large-scale scenario using the same 100M dataset. However, we filter by text to obtain *noisy labels* and follow an *inductive* approach by training a classifier for novel classes, such that the 100M collection is not needed at inference.

**Graph neural networks** are generalizations of convolutional networks to non-Euclidean spaces (Bronstein et al., 2017). Early *spectral methods* (Bruna et al., 2014; Henaff et al., 2015) have been succeeded by *Chebyshev polynomial* approximations (Defferrard et al., 2016), which avoid the high computational cost of computing eigenvectors. *Graph convolutional networks* (GCN) (Kipf & Welling, 2017) provide a further simplification by a *first-order* approximation of graph filtering and are applied to *semi-supervised* (Kipf & Welling, 2017) and subsequently to *few-shot learning* (Garcia & Bruna, 2018). In Kipf & Welling (2017), the loss function is applied to labeled examples to make predictions on unlabeled ones. Similarly in Garcia & Bruna (2018), GCNs make predictions on novel class examples. Gidaris & Komodakis (2019) use Graph Neural Networks as denoising autoencoders to generate class weights for novel classes. In contrast, we cast GCNs as *binary classifiers* discriminating clean from noisy examples: we apply a loss function to all examples, and then use the inferred probabilities as a class relevance measure, effectively cleaning the data.

Our counter-intuitive objective of treating all noisy examples as negative can be compared to treating each example as a different class in *instance-level discrimination* (Wu et al., 2018). In fact, our loss function is similar to *noise-contrastive estimation* (NCE) (Gutmann & Hyvärinen, 2010) used in that work. According to our experiments, our GCN-based classifier outperforms classical LP (Zhou et al., 2003a) used for a similar purpose by Rohrbach et al. (2013).

## 3 PROBLEM FORMULATION

We consider a space $\mathcal{X}$ of examples. We are given a set $X_{\mathcal{L}} \subset \mathcal{X}$ of examples, each having a *clean* (manually verified) label in a set $C_{\mathcal{L}}$ of classes with $|C_{\mathcal{L}}| = K_{\mathcal{L}}$. For any set $X \subset \mathcal{X}$, we denote by $X^c$ its subset of examples having a label in class $c$. We assume that the number $|X_{\mathcal{L}}^c|$ of examples labeled in each class $c \in C_{\mathcal{L}}$ is only $k$, typically in $\{1, 2, 5, 10, 20\}$. We are also given an additional set $X_{\mathcal{Z}}^c$ of examples, each with a set of *noisy* labels in $C_{\mathcal{L}}$. The *extended* set of examples for class $c$ is now $X_{\mathcal{E}}^c = X_{\mathcal{L}}^c \cup X_{\mathcal{Z}}^c$. Examples or sets of examples having clean (noisy) labels are referred to as clean (noisy) as well. The goal is to train a classifier, using the additional noisy set in order to improve the accuracy compared to only using the small clean set.

We assume that we are given a feature extractor $g_\theta : \mathcal{X} \to \mathbb{R}^d$, mapping an example to a $d$-dimensional vector. For instance, when examples are images, the feature extractor is typically a *convolutional neural network* (CNN) and $\theta$ are the parameters of all layers.

In this work, we assume that the noisy set $X_{\mathcal{Z}}$ is collected via web crawling with examples that are images accompanied with free-form text description and/or user tags originating from community photo collections. To make use of text data, we assume that the names of classes in $C_{\mathcal{L}}$ are given. An example in $X_{\mathcal{Z}}$ is given a label in class $c \in C_{\mathcal{L}}$ if its textual information contains the name of class $c$; it may then have none, one or more labels. In this way, we automatically infer labels for $X_{\mathcal{Z}}$ without human effort, which are however *noisy*.

## 4 CLEANING WITH GRAPH CONVOLUTIONAL NETWORKS

We perform cleaning by predicting a *class relevance* measure for each noisy example in $X_{\mathcal{Z}}^c$, independently per class $c \in C_{\mathcal{L}}$. To simplify notation, we drop superscript $c$ where possible in this subsection and we denote $X_{\mathcal{E}}^c$ by $\{x_1, \ldots, x_k, x_{k+1}, \ldots, x_N\}$, where $X_{\mathcal{L}}^c = \{x_1, \ldots, x_k\}$ and $X_{\mathcal{Z}}^c = \{x_{k+1}, \ldots, x_N\}$. The features of these examples are similarly represented by matrix $V = [\mathbf{v}_1, \ldots, \mathbf{v}_k, \mathbf{v}_{k+1}, \ldots, \mathbf{v}_N] \in \mathbb{R}^{d \times N}$, where $\mathbf{v}_i = g_\theta(x_i)$ for $i = 1, \ldots, N$.

We construct an affinity matrix $A \in \mathbb{R}^{N \times N}$ with elements $a_{ij} = [\mathbf{v}_i^\top \mathbf{v}_j]_+$ if examples $\mathbf{v}_i$ and $\mathbf{v}_j$ are reciprocal nearest neighbors in $X_{\mathcal{E}}^c$ and 0 otherwise. Matrix $A$ has zero diagonal, but self-connections are added and then $A$ is normalized as $\tilde{A} = D^{-1}(A + I)$ with $D = \text{diag}((A + I)\mathbf{1})$ being the degree matrix of $A + I$ and $\mathbf{1}$ the all-ones vector.

*Graph convolutional networks* (GCNs) (Kipf & Welling, 2017) are formed by a sequence of layers. Each layer is a function $f_\Theta : \mathbb{R}^{N \times N} \times \mathbb{R}^{l \times N} \to \mathbb{R}^{n \times N}$ of the form

$$f_\Theta(\tilde{A}, Z) = h(\Theta^\top Z \tilde{A}), \tag{1}$$

where $Z \in \mathbb{R}^{l \times N}$ represents the input features, $\Theta \in \mathbb{R}^{l \times n}$ holds the parameters of the layer to be learned, and $h$ is a nonlinear activation function. Function $f_\Theta$ maps $l$-dimensional input features to $n$-dimensional output features.

In this work we consider a two layer GCN with a scalar output per example. This network is a function $F_\Theta : \mathbb{R}^{N \times N} \times \mathbb{R}^{d \times N} \to \mathbb{R}^N$ given by

$$F_\Theta(\tilde{A}, V) = \sigma(\Theta_2^\top [\Theta_1^\top V \tilde{A}]_+ \tilde{A}), \tag{2}$$

where $\Theta = \{\Theta_1, \Theta_2\}$, $\Theta_1 \in \mathbb{R}^{d \times m}$, $\Theta_2 \in \mathbb{R}^{m \times 1}$, $[\cdot]_+$ is the positive part or ReLU function (Nair & Hinton, 2010) and $\sigma(x) = (1 + e^{-x})^{-1}$ for $x \in \mathbb{R}$ is the sigmoid function. Function $F_\Theta$ performs feature propagation through the affinity matrix in an analogy to classical graph-based propagation methods for classification (Zhou et al., 2003a) or search (Zhou et al., 2003b).

The output $F_\Theta(\tilde{A}, V)$ is a vector of length $N$, with element $F_\Theta(\tilde{A}, V)_i$ in $[0, 1]$ representing a relevance value of example $x_i$ for class $c$. To learn the parameters $\Theta$, we treat the GCN as a *binary classifier* where target output 1 corresponds to clean examples and 0 to noisy. In particular, we minimize the loss function

$$L_{\mathcal{G}}(V, \tilde{A}; \Theta) = -\frac{1}{k} \sum_{i=1}^{k} \log\left(F_\Theta(\tilde{A}, V)_i\right) - \frac{\lambda}{N - k} \sum_{i=k+1}^{N} \log\left(1 - F_\Theta(\tilde{A}, V)_i\right). \tag{3}$$

This is a binary cross-entropy loss function where noisy examples are given an importance weight $\lambda$. Given the propagation on the nearest neighbor graph, and depending on the relative importance $\lambda$ of the second term, noisy examples that are strongly connected to clean ones are still expected to receive high class relevance, while noisy examples that are not relevant to the current class are expected to get a class relevance near zero.

The impact of parameter $\lambda$ is validated in Section 6, where we show that the fewer the available clean images are (smaller $k$) the smaller the importance weight should be. As is standard practice for GCNs in classification (Kipf & Welling, 2017), training is performed in batches of size $N$, that is the entire set of features.

Figure 2 shows examples of clean images, corresponding noisy ones and the predicted relevance. Thanks to the visual similarity to the clean image, we can use relevance to resolve cases of polysemy, *e.g. black widow (spider) vs. black widow (superhero)*, or cases like *pineapple vs. pineapple juice*.

**Discussion.** Loss function (3) is similar to *noise-contrastive estimation* (NCE) (Gutmann & Hyvärinen, 2010) as used by Wu et al. (2018) for *instance-level discrimination*, whereas we discriminate clean from noisy examples. The semi-supervised learning setup of GCNs (Kipf & Welling, 2017) uses a loss function that applies only to the labeled examples, and makes discrete predictions on unlabeled examples. In our case, all examples contribute to the loss but with different importance, while we infer real-valued class relevance for the noisy examples, to be used for subsequent learning.

Function $F_\Theta$ in (2) reduces to a Multi-Layer Perceptron (MLP) when the affinity matrix $A$ is zero, in which case all examples are disconnected. Using an MLP to perform cleaning would take each

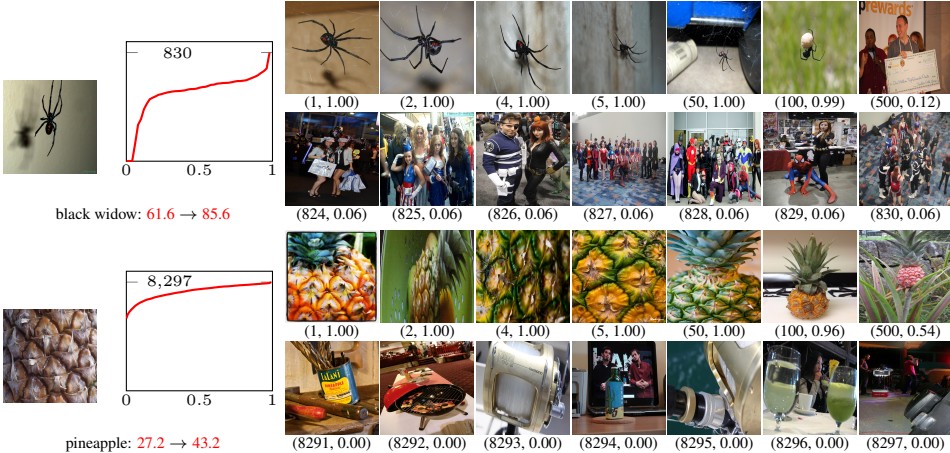

Figure 2: Examples of clean images (left) for 1-shot classification, cumulative histogram of the predicted relevance for noisy images (middle), and representative noisy images (right), each having its position in the (descending) ranked list according to relevance and relevance value reported below. Test accuracy without and with additional data using class prototypes (6) is shown next to class names.

example into account independently of the others, while the GCN considers the collection of examples as a whole. MLP training is performed identically to GCN by minimizing (3). We compare the two alternatives in our experiments.

## 5 LEARNING A CLASSIFIER WITH FEW CLEAN AND MANY NOISY EXAMPLES

Our cleaning process applies when the clean labeled examples are few, but assumes a feature extractor $g_\theta$. That is, representation learning, label cleaning and classifier learning are decoupled. We follow few-shot learning in that we learn the representation by supervised classification on a set of *base classes*, obtaining $g_\theta$, and then solving new classification tasks on a distinct set of *novel classes*. In these new tasks, we assume few clean and many noisy labels as specified in Section 3, perform GCN-based cleaning as described in Section 4, and learn a classifier by weighing examples according to class relevance. Representation and classifier learning are described below.

### 5.1 COSINE-SIMILARITY BASED CLASSIFIER

We use a *cosine-similarity based classifier* (Gidaris & Komodakis, 2018; Qi et al., 2018), or *cosine classifier* for short. Given classes $C$ with $|C| = K$, each class $c \in C$ is represented by a learnable parameter $\mathbf{w}_c \in \mathbb{R}^d$. The *prediction* of example $x \in \mathcal{X}$ is the class $c$ of maximum cosine similarity $\hat{\mathbf{w}}_c^\top \hat{g}_\theta(x)$[1]

$$\pi_{\theta, W}(x) = \arg\max_c \hat{\mathbf{w}}_c^\top \hat{g}_\theta(x), \tag{4}$$

where $W = [\mathbf{w}_1, \ldots, \mathbf{w}_K] \in \mathbb{R}^{d \times K}$.

### 5.2 REPRESENTATION LEARNING: BASE CLASSES

We are given a set $X_\mathcal{B} \subset \mathcal{X}$ of examples, each having a clean label in a set of *base classes* $C_\mathcal{B}$ with $|C_\mathcal{B}| = K_\mathcal{B}$. These data are used to learn a feature representation, *i.e.* a feature extractor $g_\theta$, by learning a $K_\mathcal{B}$-way base-class classifier for unseen data in $\mathcal{X}$. The parameters $\theta$ of the feature extractor and $W_\mathcal{B}$ of the classifier are jointly learned by minimizing the cross entropy loss

$$L_\mathcal{B}(C_\mathcal{B}, X_\mathcal{B}; \theta, W_\mathcal{B}) = -\sum_{c \in C_\mathcal{B}} \frac{1}{|X_\mathcal{B}^c|} \sum_{x \in X_\mathcal{B}^c} \log(\boldsymbol{\sigma}(s \hat{W}_\mathcal{B}^\top \hat{g}_\theta(x))_c), \tag{5}$$

where $\boldsymbol{\sigma} : \mathbb{R}^K \to \mathbb{R}^K$ is the softmax function with $\boldsymbol{\sigma}(\mathbf{a})_c = e^{a_c} / \sum_{j \in C} e^{a_j}$ for $\mathbf{a} \in \mathbb{R}^K$, $s$ is a learnable *scale parameter* and $\hat{W}_\mathcal{B} = [\hat{\mathbf{w}}_1, \ldots, \hat{\mathbf{w}}_{K_\mathcal{B}}] \in \mathbb{R}^{d \times K_\mathcal{B}}$. Learning and inference are

---

[1]We denote the $\ell_2$-normalized counterpart of vector $\mathbf{x}$ by $\hat{\mathbf{x}}$. Similarly, if $\mathbf{y} = f(x)$, denote $\hat{\mathbf{y}}$ by $\hat{f}(x)$.

performed on base classes by $L_{\mathcal{B}}(C_{\mathcal{B}}, X_{\mathcal{B}}; \theta, W_{\mathcal{B}})$ (5) and $\pi_{\theta, W_{\mathcal{B}}}$ (4), respectively. As a result, learned feature extractor parameters $\theta$ are used for base or novel classes, while the classifier parameters $W_{\mathcal{B}}$ can be used for base class or *all-class* classification, as discussed below.

### 5.3 New classification tasks: novel or all classes

Each new task is related to a set of *novel classes* $C_{\mathcal{L}}$, disjoint from $C_{\mathcal{B}}$. The goal is to learn a $K_{\mathcal{L}}$-way novel-class classifier or a $K_{\mathcal{A}}$-way classifier on all classes $C_{\mathcal{A}} = C_{\mathcal{B}} \cup C_{\mathcal{L}}$ for unseen data in $\mathcal{X}$, where $K_{\mathcal{A}} = K_{\mathcal{B}} + K_{\mathcal{L}}$. Unlike the typical few-shot learning task, each novel class contains few clean and many noise examples.

Prior to learning classifiers for novel classes, training examples $x_i \in X_{\mathcal{Z}}^c$ are weighted by their relevance $r(x_i)$ to class $c$. For a noisy example $x_i \in X_{\mathcal{E}}^c$, we define $r(x_i) = F_{\Theta}(\tilde{A}, V)_i$ where $F_{\Theta}(\tilde{A}, V)$ is the output vector of the GCN, while for a clean example $x_i \in X_{\mathcal{L}}^c$ we fix $r(x_i) = 1$. Note that optimizing (3) does not guarantee $F_{\Theta}(\tilde{A}, V)_i = 1$ for clean examples $x_i \in X_{\mathcal{L}}^c$. We define $r(X) = \sum_{x \in X} r(x)$ for any set $X \subset \mathcal{X}$.

We first assume that we no longer have access to examples of base classes in new classification tasks and consider two different classifiers, class prototypes and cosine-similarity based classifier. Then, this assumption is dropped and the classifier and feature representation are learned jointly by fine-tuning the entire network.

**Class prototypes.** For each novel class $c \in C_{\mathcal{L}}$, we define *prototype* $\mathbf{w}_c$ by

$$\mathbf{w}_c = \frac{1}{r(X_{\mathcal{E}}^c)} \sum_{x \in X_{\mathcal{E}}^c} r(x) g_{\theta}(x). \tag{6}$$

Prototypes are fixed vectors, not learnable parameters. Collecting them into matrix $W_{\mathcal{L}} = [\mathbf{w}_1, \ldots, \mathbf{w}_{K_{\mathcal{L}}}] \in \mathbb{R}^{d \times K_{\mathcal{L}}}$, $K_{\mathcal{L}}$-way prediction on novel classes is made by classifier $\pi_{\theta, W_{\mathcal{L}}}$ (4), while $K_{\mathcal{A}}$-way prediction on all (base and novel) classes by $\pi_{\theta, W_{\mathcal{A}}}$, where $W_{\mathcal{A}} = [W_{\mathcal{B}}, W_{\mathcal{L}}]$ and $W_{\mathcal{B}}$ is learned according to $L_{\mathcal{B}}(C_{\mathcal{B}}, X_{\mathcal{B}}; \theta, W_{\mathcal{B}})$ (5) and then kept fixed.

**Cosine classifier learning.** Similarly to Section 5.2, given clean and noisy novel-class examples $X_{\mathcal{E}}$, we learn a parametric cosine classifier with parameters $W_{\mathcal{L}} = [\mathbf{w}_1, \ldots, \mathbf{w}_{K_{\mathcal{L}}}] \in \mathbb{R}^{d \times K_{\mathcal{L}}}$ by minimizing the weighted cross entropy loss $L_{\mathcal{L}}(C_{\mathcal{L}}, X_{\mathcal{E}}; \theta, W_{\mathcal{L}})$ over $W_{\mathcal{L}}$, where

$$L_{\mathcal{L}}(C_{\mathcal{L}}, X_{\mathcal{E}}; \theta, W_{\mathcal{L}}) = -\sum_{c \in C_{\mathcal{L}}} \frac{1}{r(X_{\mathcal{E}}^c)} \sum_{x \in X_{\mathcal{E}}^c} r(x) \log(\boldsymbol{\sigma}(s \hat{W}_{\mathcal{L}}^{\top} \hat{g}_{\theta}(x))_c), \tag{7}$$

while the parameters $\theta$ of the feature extractor are fixed. The scale parameter $s$ is also fixed to the value obtained during base class learning. Prediction on novel only or all classes is then made as in the previous case.

**Deep network fine-tuning.** We now drop the assumption that base class examples are not accessible and, given all examples $X_{\mathcal{A}} = X_{\mathcal{B}} \cup X_{\mathcal{E}}$, we jointly learn the parameters $\theta$ of the feature extractor and $W_{\mathcal{A}} = (W_{\mathcal{B}}, W_{\mathcal{L}})$ of the $K_{\mathcal{A}}$-way cosine classifier for all classes by minimizing loss function

$$L_{\mathcal{A}}(C_{\mathcal{A}}, X_{\mathcal{A}}; \theta, W_{\mathcal{A}}) = L_{\mathcal{B}}(C_{\mathcal{B}}, X_{\mathcal{B}}; \theta, W_{\mathcal{B}}) + L_{\mathcal{L}}(C_{\mathcal{L}}, X_{\mathcal{E}}; \theta, W_{\mathcal{L}}). \tag{8}$$

Note that, due to overfitting on the few available examples, such learning is avoided in a few-shot learning setup. In a few cases, it takes the form of fine-tuning including all base class data (Qi et al., 2018), or only lasts for a few iterations when the base class data is not accessible (Finn et al., 2017).

## 6 Experiments

### 6.1 Experimental setup

**Datasets and task setup.** We extend the *Low-Shot ImageNet benchmark* introduced by Hariharan & Girshick (2017) by assuming many noisy examples for novel classes, in addition to the few clean ones. In this benchmark, the 1000 ImageNet classes (Russakovsky et al., 2015) are split into 389

base classes and 611 novel classes. The validation set contains 193 base and 300 novel classes, and the test set the remaining 196 base and 311 novel classes. The standard benchmark includes $k$-shot classification, *i.e.* classification on $k$ clean examples per class, which we extend to $k$ clean and many noisy examples per class, with $k \in \{1, 2, 5, 10, 20\}$. Similar to Hariharan & Girshick (2017) we perform 5 tasks, each drawing a subset of $k$ clean examples per class. We report the average top-5 accuracy over the 5 tasks on novel or all classes of the test set.

We use the YFCC100M dataset (Thomee et al., 2016) as a source for additional data with noisy labels. It contains approximatively 100M images collected from Flickr. Each image comes with a text description obtained from the user title and caption. We use the text description to obtain images with noisy labels. as discussed in Section 3. This process results in very different numbers of additional examples per class, with a minimum of zero for classes *maillot* and *missile*, and a maximum of 620,142 for class *church/church building*.

**Representation and classifier learning.** In most experiments, we use ResNet-10 (He et al., 2016) as feature extractor as in Gidaris & Komodakis (2018). Classification for novel classes is performed with *class prototypes* (6), *cosine classifier learning* (7) or *deep network fine-tuning* (8). Hyper-parameters such as batch size and number of epochs, are tuned on the validation set. Possible values are 2048, 4096, and 8192 for batchsize and 10, 30 and 50 for number of epochs. The learning rate starts from 0.1 and is reduced to 0.001 at the end of training with *cosine annealing* (Loshchilov & Hutter, 2017).

We handle the imbalance of the noisy set by normalizing by $r(X_c)$ in (7). Prototypes (6) are used to initialize $W_\mathcal{L}$ of cosine classifier in (7), and the learned $W_\mathcal{L}$ is used to initialize the corresponding part of $W_\mathcal{A}$ when fine-tuning the network by (8). In the latter case, we train all layers for 10 epochs with learning rate 0.01. We ignore examples $x_i$ with relevance $r(x_i) < 0.1$ to reduce the complexity when fine-tuning the network.

We also report results with ResNet-50 as feature extractor, using the model trained on base classes by Hariharan & Girshick (2017). Following Douze et al. (2018), we apply PCA to the features to reduce their dimensionality to 256. Base classes are represented by class prototypes (6) in this case.

GCN training is performed with Adam optimizer and a learning rate of 0.1 for 100 iterations. We use dropout with probability 0.5. The dimensionality of the input descriptors is $d = 512$ for ResNet-10 and $d = 256$ for ResNet-50 (after PCA). Dimensionality of the internal representation in (1) is $m = 16$. The affinity matrix is constructed with reciprocal top-50 nearest neighbors.

**Baselines.** We implement and evaluate several baseline methods. $\beta$-*cleaning* assigns $r(x_i) = \beta$ to all additional examples. We report results for $\beta = 1.0$ (unit relevance score) and $\beta^*$, the optimal $\beta$ for all $k$ obtained on the validation set. *MLP*, discussed in Section 4, learns a nonlinear mapping to assign relevance, but does not propagate over the graph. *Label Propagation* (LP) (Zhou et al., 2003a) propagates information by a linear operation. It solves the linear system $(I - \alpha D^{-1/2} A D^{-1/2})\mathbf{r}_c = \mathbf{y}_c$ (Iscen et al., 2017) for each class $c$, where $D$ is the degree matrix of $A$, $\alpha = 0.9$ and $\mathbf{y}_c \in \mathbb{R}^N$ is a $k$-hot binary vector indicating the labeled examples of class $c$. Relevance $r(x_i)$ is then the $i$-th element $(\mathbf{r}_c)_i$ of the solution.

## 6.2 EXPERIMENTAL RESULTS

**The impact of importance weight** $\lambda$ is measured on the validation set and the best performing value is used on the test set for each value of $k$. Results are shown in Appendix A. The larger the value of $\lambda$, the more the loss encourages noisy examples to be classified as negatives. As a consequence, large (small) $\lambda$ results in smaller (larger) relevance, on average, for noisy examples. The optimal $\lambda$ per value of $k$ suggests that the fewer the clean examples the larger the need for additional ones.

**Comparison with baselines using additional data** is presented in Table 1. The use of additional data is mostly harmful for $\beta$-weighting except for 1 and 2-shot. MLP offers improvements in most cases, implying that it manages to appropriately downweigh irrelevant examples. The consistent improvement of our method compared to MLP, especially large for small $k$, suggests that it is beneficial to incorporate relations, with the affinity matrix $A$ modeling the structure of the feature space. LP is a classic approach that also uses $A$ but is a linear operation with no parameters, and is inferior to our method. The gain of cleaning ($\beta = 1$ *vs.* ours) ranges from 11% to 20%.

| Method | $k$=1 | 2 | 5 | 10 | 20 |
|---|---|---|---|---|---|
| FEW CLEAN EXAMPLES | | | | | |
| Class proto. Gidaris & Komodakis (2018) | 45.3±0.65 | 57.1±0.37 | 69.3±0.32 | 74.8±0.20 | 77.8±0.24 |
| FEW CLEAN & MANY NOISY EXAMPLES | | | | | |
| $\beta$-weighting, $\beta = 1$ | 56.1±0.06 | 56.4±0.08 | 57.1±0.05 | 57.7±0.08 | 58.7±0.06 |
| $\beta$-weighting, $\beta^*$ | 55.6±0.24 | 58.3±0.14 | 63.4±0.25 | 67.5±0.34 | 71.0±0.22 |
| Label Propagation Zhou et al. (2003a) | 62.6±0.35 | 67.0±0.41 | 74.6±0.30 | 76.3±0.23 | 77.7±0.18 |
| MLP | 63.6±0.41 | 68.8±0.42 | 73.9±0.25 | 75.6±0.21 | 77.6±0.21 |
| Ours | **67.8±0.10** | **70.9±0.30** | **73.7±0.17** | **76.1±0.12** | **78.2±0.14** |

Table 1: Comparison with baselines using noisy examples. We report top-5 accuracy on novel classes with classification by class prototypes (6).

| METHOD | NOVEL CLASSES | | | | | ALL CLASSES | | | | |
|---|---|---|---|---|---|---|---|---|---|---|
| | $k$=1 | 2 | 5 | 10 | 20 | $k$=1 | 2 | 5 | 10 | 20 |
| RESNET-10 – FEW CLEAN EXAMPLES | | | | | | | | | | |
| Proto.-Nets (Snell et al., 2017) | 39.3 | 54.4 | 66.3 | 71.2 | 73.9 | 49.5 | 61.0 | 69.7 | 72.9 | 74.6 |
| Logistic reg. w/ H (Wang et al., 2018b) | 40.7 | 50.8 | 62.0 | 69.3 | 76.5 | 52.2 | 59.4 | 67.6 | 72.8 | 76.9 |
| PMN w/ H (Wang et al., 2018b) | 45.8 | 57.8 | 69.0 | 74.3 | 77.4 | 57.6 | 64.7 | 71.9 | 75.2 | 77.5 |
| Class proto. (Gidaris & Komodakis, 2018) | 45.3±0.65 | 57.1±0.37 | 69.3±0.32 | 74.8±0.20 | 77.8±0.24 | 57.0±0.36 | 64.7±0.16 | 72.5±0.18 | 75.8±0.16 | 77.4±0.19 |
| Class proto. w/ Att. (Gidaris & Komodakis, 2018) | 45.8±0.74 | 57.4±0.38 | 69.6±0.27 | 75.0±0.29 | 78.2±0.23 | 58.1±0.48 | 65.2±0.15 | 72.9±0.25 | 76.6±0.18 | 78.8±0.16 |
| RESNET-10 – FEW CLEAN & MANY NOISY EXAMPLES | | | | | | | | | | |
| Ours - class proto. (6) | **67.8±0.10** | **70.9±0.30** | **73.7±0.20** | **76.1±0.16** | **78.2±0.14** | **70.3±0.05** | **72.1±0.18** | **74.1±0.12** | **75.6±0.13** | **76.9±0.09** |
| Ours - cosine (7) | **73.2±0.14** | **75.3±0.25** | **75.6±0.24** | **78.5±0.32** | **80.7±0.26** | **71.9±0.07** | **74.0±0.23** | **76.5±0.16** | **78.3±0.23** | **80.2±0.18** |
| Ours - fine-tune (8) | **74.6±0.13** | **76.6±0.26** | **78.2±0.23** | **80.9±0.34** | **82.9±0.20** | **76.0±0.10** | **77.3±0.13** | **78.7±0.19** | **80.7±0.25** | **82.2±0.14** |
| RESNET-50 – FEW CLEAN EXAMPLES | | | | | | | | | | |
| Proto.-Nets (Snell et al., 2017) | 49.6 | 64.0 | 74.4 | 78.1 | 80.0 | 61.4 | 71.4 | 78.0 | 80.0 | 81.1 |
| PMN w/ H (Wang et al., 2018b) | 54.7 | 66.8 | 77.4 | 81.4 | 83.8 | 65.7 | 73.5 | 80.2 | 82.8 | 84.5 |
| RESNET-50 – FEW CLEAN & MANY UNLABELED EXAMPLES | | | | | | | | | | |
| Diffusion (Douze et al., 2018) | 63.6±0.61 | 69.5±0.60 | 75.2±0.40 | 78.5±0.34 | 80.8±0.18 | - | - | - | - | - |
| Diffusion - logistic (Douze et al., 2018) | 64.0±0.70 | 71.1±0.82 | 79.7±0.38 | 83.9±0.10 | 86.3±0.17 | - | - | - | - | - |
| RESNET-50 – FEW CLEAN & MANY NOISY EXAMPLES | | | | | | | | | | |
| Ours - class proto. (6) | **69.7±0.44** | **73.7±0.56** | **77.0±0.20** | **79.9±0.30** | **81.9±0.29** | **73.8±0.33** | **76.6±0.36** | **78.9±0.19** | **80.8±0.21** | **82.2±0.14** |
| Ours - cosine (7) | **78.0±0.38** | **80.2±0.33** | **80.9±0.17** | **83.7±0.19** | **85.7±0.11** | **77.6±0.26** | **79.1±0.20** | **79.9±0.09** | **82.1±0.22** | **83.8±0.11** |
| Ours - fine-tune (8) | **80.8±0.25** | **83.0±0.23** | **83.8±0.39** | **86.4±0.23** | **88.5±0.20** | **81.6±0.20** | **83.2±0.16** | **84.3±0.23** | **86.2±0.17** | **87.8±0.03** |

Table 2: Comparison to the state of the art on the Low-shot ImageNet benchmark. We report top-5 accuracy on novel and all classes. We use class prototypes (6), cosine classifier learning (7) and deep network fine-tuning (8) for classification with our GCN-based data addition method.

**Comparison with the state of the art** is presented in Table 2. We significantly improve the performance by using additional data and cleaning compared to a number of different approaches, including the work by Gidaris & Komodakis (2018), which is our starting point. As expected, the gain is more pronounced for small $k$, reaching more than 20% improvement for 1-shot novel accuracy.

Closest to ours is the work of Douze et al. (2018), who use the same experimental setup and the same additional data, but without filtering by text and using noisy labels. We outperform their approach in all cases, while requiring much less computation: *offline*, we construct a separate small graph per class rather than a single graph over the entire 100M collection; *online*, we perform inference by cosine similarity to one prototype per class or a learned classifier rather than iterative diffusion on the entire collection. Note that by ignoring examples that are not given any noisy label, we are only using a tiny fraction of the 100M collection: in particular, only 3,744,994 images for the 311-class test split of the Low-shot ImageNet benchmark. In contrast to Douze et al. (2018), additional data brings improvement even at 20-shot with classifier learning or network fine-tuning. Most importantly, our approach does not require the entire 100M collection at inference.

# 7 CONCLUSIONS

In this paper we have introduced a new method for assigning class relevance to noisy images obtained by textual queries with class names. Our approach leverages one or a few labeled images per class and relies on a graph convolutional network (GCN) to propagate visual information from the labeled images to the noisy ones. The GCN is a binary classifier discriminating clean from noisy examples using a weighted binary cross-entropy loss function and inferring "clean" probability as a relevance measure for that class. Experimental results show that using noisy images weighted by this relevance measure significantly improves the classification accuracy.

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

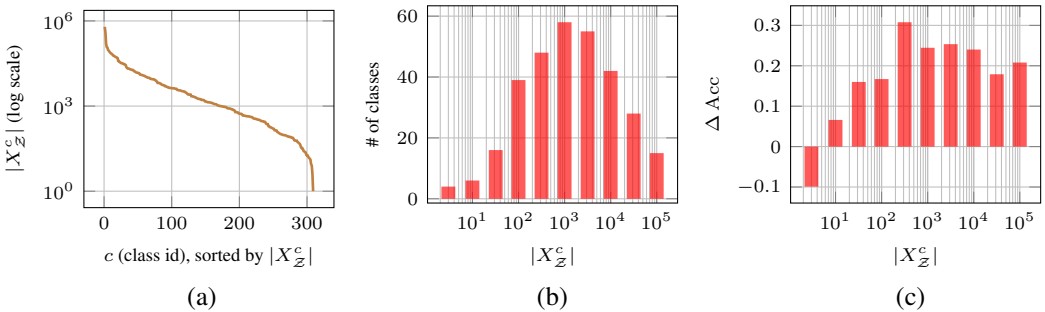

|   |   |   |
|---|---|---|
| (a) | (b) | (c) |

Figure 3: (a) Number of additional images per class $c$ sampled from YFCC-100M for all novel classes of Low-Shot ImageNet. (b) Number of classes per group, when $|X_{\mathcal{Z}}^c|$ is sampled logarithmically into groups. (c) Accuracy improvement $\Delta$ Acc (difference of accuracy between our method with noisy examples and the baseline without noisy examples) for prototype classifier, for same groups as in (b).

## A  APPENDIX

**Noisy data statistics.** We present statistics about the noisy examples of novel classes and the improvements of our method per class. Figure 3 (a) shows that the noisy examples for novel classes are long tailed (in log scale). There is a significant number of classes where we end up with less than 1000 extra examples, but we improve nevertheless; see Figure 3 (c). A small exception is 4 very rare classes out of 311, with around 3 additional images per class (leftmost bin in Figure 3 (b) and (c)). Note that in real world applications, one could use more resources like web queries for additional data.

**Impact of importance weight $\lambda$.** We present the impact of $\lambda$ (3) for different values of $k$ in the validation set of the extended Low-shot ImageNet benchmark in Figure 4.

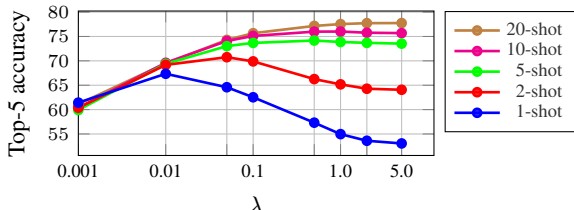

Figure 4: Impact of $\lambda$ on the validation set of the extended Low-shot ImageNet benchmark with YFCC-100M for noisy examples using class prototypes (6).

