# OpenReview forum: "Graph convolutional networks for learning with few clean and many noisy labels"
_ICLR.cc/2020/Conference — Reject_

### Official Review · AnonReviewer3 · 2019-10-23
**Official Blind Review #3**

**Rating:** 6

**Review:**

This paper studies the problem of learning from multiple tasks and additional noisy data. The proposed representation learning method first assigns each noisy data a relevance score using the topological information. Then the authors propose to minimize a combination of the loss of a class-prototype learning loss and a cosine classifier learning loss to learn a good representation generator g_theta. The empirical study validates the effectiveness of the proposed method.

I have the following comments,

1. The studied problem that learning from few-shot data and large-scale noisy data is interesting. According to the experimental results, the proposed method seems to be promising.

2. The learning procedure is confusing. It is highly recommended to provide the pseudocode of the proposed method.

3. Since there are many tasks and each task has a large-scale data, I'm afraid that the running time will explode. How to deal with this issue?

**Experience Assessment:**

I have read many papers in this area.

**Review Assessment: Checking Correctness Of Derivations And Theory:**

I carefully checked the derivations and theory.

**Review Assessment: Checking Correctness Of Experiments:**

I carefully checked the experiments.

**Review Assessment: Thoroughness In Paper Reading:**

I read the paper at least twice and used my best judgement in assessing the paper.

---

> ### Author Response · Authors · 2019-11-09
> **Response to Reviewer #3**
>
> We would like to thank the reviewer for the positive feedback. We reply to the the two questions below.
>
> Q1: The learning procedure is confusing. It is highly recommended to provide the pseudocode of the proposed method.
>
> R1: We will provide the pseudocode in the future versions of the paper:
>
> Training:
>
> X_L : clean set
> C_L : class set
> X_Z : noisy set
>
> # For each class name
> For c in C_L:
>
> 	#Take the clean examples belonging to this class
> 	X_L^c : subset of X_L with label c
>
> 	#Only consider noisy examples with the class name in the text
> 	X_Z^c = filter_by_text(X_Z)
>
> 	# Build the graph for this class, and learn the GCN for cleaning
> 	A^c = build_graph(X_Z^c)
> 	M^c = GCN_model(X_L^c, X_Z^c, A^c)
>
> 	#Clean examples always get weight 1
> 	for i in X_L^c:
> 		r_i = 1.0
>
> 	#Noisy examples get the learned weight
> 	for i in X_Z^c:
> 		r_i = assign_relevance(M^c(X_Z^c(i))
>
> 	#Add the noisy examples to the list of training images for this class
> 	X_L^c = concatenate(X_L^c, X_z^c)
>
> #Learn a classifier jointly for all classes. Use the relevance weights for noisy examples when learning the classifier
> W = train_classifier(X_L^c, r)
>
> Testing
>
> Given test image Q
>
> v = extract_feature(Q)
> scores = W^T v
> prediction = argmax(scores)
>
>
> Q2: Since there are many tasks and each task has a large-scale data, I'm afraid that the running time will explode. How to deal with this issue?
>
> R2: The complexity is linear in the number of classes, since classes are processed independently. Furthermore, text filtering is applied before cleaning, which reduces the number of images to be considered for a given class. Please also see the response R1 to reviewer1.

---

### Official Review · AnonReviewer1 · 2019-10-26
**Official Blind Review #1**

**Rating:** 6

**Review:**

This paper presents a classification method when the data consists of few clean labels and many noisy labels. The authors propose to construct a graph structure within each class and use graph convolutional network to determine the clean/noisy labels of samples in each class. The model is based on a binary cross entropy loss function in each class, which learns the probability of labels to be clean. And such "clean" probability is used as the measure of relevance score between the sample different classes.

The idea of this paper is straightforward and the experimental results seem promising. The authors compare with several related methods and show the proposed method has better performance in few shot learning experiments.

For the motivation of this methods, why would the graph be constructed within each class? If there is correlation between different classes, how could the model use such class-wise correlation to clean the label?

Maybe I missed it, but how is the relevance score / predicted label determined for testing data given the graphs constructed in each class of training data?

**Experience Assessment:**

I do not know much about this area.

**Review Assessment: Checking Correctness Of Derivations And Theory:**

I did not assess the derivations or theory.

**Review Assessment: Checking Correctness Of Experiments:**

I assessed the sensibility of the experiments.

**Review Assessment: Thoroughness In Paper Reading:**

I read the paper at least twice and used my best judgement in assessing the paper.

---

> ### Author Response · Authors · 2019-11-09
> **Response to Reviewer #1**
>
> We would like to thank the reviewer for the positive feedback. We reply to the the two questions below.
>
> Q1: For the motivation of this method, why would the graph be constructed within each class? If there is a correlation between different classes, how could the model use such class-wise correlation to clean the label?
>
> R1: The most general graph would be constructed based on image and text similarities combined. Here, we pre-filter with text similarity, i.e., label names, and then build the graph based on visual similarities. This permits (a) to significantly reduce the size of the graph and hence the complexity and (b) to reduce the noise during the cleaning task. We agree that operating on the more complex graph could be the subject of future research, but a significantly different method would be required and the gain of the correlation is not granted.
>
>
> Q2: Maybe I missed it, but how is the relevance score / predicted label determined for testing data given the graphs constructed in each class of training data?
>
> R2: There is no relevance score assigned to the test data. Relevance scores are only used during training. In particular, we build per-class graphs using the training data, assign each training example a relevance score (Section 4), and train a classifier using the training data and the corresponding relevance scores (Section 5). Now given a test image, a prediction is simply made by the classifier; no data or relevance scores are used. See also pseudo-code in response R1 to reviewer 3.

---

### Official Review · AnonReviewer2 · 2019-11-10
**Official Blind Review #2**

**Rating:** 6

**Review:**

The paper makes a significant attempt at solving one of the practical problems in machine learning -- learning from many noisy and limited number of clean labels. This setting is presumably more practical than the setting of few-shot learning. Noisy labels are often abundantly available and investing in methods that can take the noise into account for building a discriminative model is quite timely.

To be honest, the theoretical contribution of the paper is limited.  The authors make use of the nearest neighbour graph obtained from a reduced-dimensional set of features to compute the weights of the noisy labels that must guide the predictive model. From this perspective, the paper seems like an application of existing tools (such as CNN, graph convolutional network and binary classification). However, that does not undermine the superior results the authors have received in the novel application they have targeted. I appreciate the effort that went validating these ideas with real-world datasets.

In future, I would like to see a joint approach to such training, where the function g(), the nearest neighbour graph loss and the classification loss are all tied in the same objective function and are optimized jointly.

The paper has few really minor grammatical errors and typos. Please fix those before uploading the final draft.

**Experience Assessment:**

I have published one or two papers in this area.

**Review Assessment: Checking Correctness Of Derivations And Theory:**

I assessed the sensibility of the derivations and theory.

**Review Assessment: Checking Correctness Of Experiments:**

I assessed the sensibility of the experiments.

**Review Assessment: Thoroughness In Paper Reading:**

I read the paper thoroughly.

---

> ### Author Response · Authors · 2019-11-10
> **Response to Reviewer #2**
>
> We thank the review for their positive comments.  We agree that our main contribution is to cast a graph convolution network as a binary classifier learning to discriminate clean from noisy data and show its excellent results for few-shot learning.
>
> Q1: In future, I would like to see a joint approach to such training, where the function g(), the nearest neighbour graph loss and the classification loss are all tied in the same objective function and are optimized jointly.
>
> R1: We fully agree that this is an interesting direction for future research, which should result in a further increase in performance.  It would be interesting to see if the feature extractor can benefit from the cleaning of noisy images during learning, resulting in more robust feature descriptors.

---

### Decision · Program_Chairs · 2019-12-19

**Decision:**

Reject

**Comment:**

The paper combines graph convolutional networks with noisy label learning. The reviewers feel that novelty in the work is limited and there is a need for further experiments and  extensions.